# Sound feature representations decorrelate across the mouse auditory pathway

**Etienne Gosselin**[1], **Sophie Bagur**[1], **Sara Jamali**[1], **Jean-Luc Puel**[2], **Jérôme Bourien**[2], **Brice Bathellier**[1]*

**1** Université Paris-Cité, Institut Pasteur, AP-HP, INSERM, Fondation pour l'Audition, Institut de l'Audition, IHU Re-Connect, Paris, France, **2** INM, Univ Montpellier, INSERM, Montpellier, France

* brice.bathellier@cnrs.fr

## Abstract

Early studies on orientation selectivity in the visual cortex have suggested that sensory systems generate new feature representations at specific processing stages. Many observations challenge this view, but in the absence of systematic, multistage measurements, the logic of how feature tuning emerges remains elusive. Here, using a generic approach based on representational similarity analysis with a noise-corrected population metric, we demonstrate in the mouse auditory system that feature representations evolve gradually with, in some cases, major, feature-specific improvements at particular stages. We observe that single frequency tuning is already fully developed in the cochlear nucleus, the first stage of processing, while tuning to higher-order features improves up to the auditory cortex, with major steps in the inferior colliculus for amplitude modulation frequency or noise bandwidth tuning and in the cortex for frequency modulation direction and for complex sound identity or direction. Moreover, we observe that intensity tuning is established in a feature-dependent manner, earlier for pure frequencies than for more complex sounds. This indicates that auditory feature computations are a mix of stepwise and gradual processes which together contribute to decorrelate sound representations.

## Introduction

The first step of sensory perception is the transformation of physical quantities in external stimuli to neural activity sent to the central nervous system. At the peripheral stage, most of the relevant information is usually encoded with high temporal precision [1–3] along a few physical dimensions such as location in retinal coordinates in the eye [4,5], frequency and intensity in the cochlea [6–8], or body location, contact magnitude and speed in tactile receptors [9,10]. By contrast, central brain areas exhibit not only a spatial organization related to peripheral representations [11–14], but also have neurons with selectivity properties that are not seen in the periphery [15,16]. The best known example of this type of transformation is the emergence of

**Data availability statement:** All datasets are freely available at https://doi.org/10.5281/zenodo.14421103, hosted by Zenodo. Custom codes used in this study are freely available at https://doi.org/10.5281/zenodo.14421103, hosted by Zenodo.

**Funding:** This work was supported by the Fondation pour l'Audition (RD-2023-1 to BB, FPA IDA02 to BB and APA 2016-03 to BB), European Research Council (ERC CoG 770841 DEEPEN to BB) Fondation pour la Recherche Médicale (SPF202005011970 to SB) The funders had no role in study design, data collection and analysis, decision to publish, or preparation of the manuscript.

**Competing interests:** The authors have declared that no competing interests exist.

**Abbreviations:** ANF, auditory nerve fiber; CN, cochlear nucleus; DCN, dorsal cochlear nucleus; IC, inferior colliculus; IHC, inner hair cell; RSA, representation similarity analysis; VCN, ventral cochlear nucleus.

contour orientation-selective neurons in the visual cortex [15]. Similarly in the auditory system, some neurons fire only for a specific range of amplitude modulation frequencies [17,18] while all auditory nerve fibers respond to all modulation frequencies [19]. Some neurons are specific to the direction of frequency modulations [20–22], while no auditory nerve fiber displays this selectivity. This feature extraction process is thought to be an essential computation for the generation of sensory representations in the brain that allow identifying objects of the external environment [23]. Therefore, precisely measuring the evolution of feature representations along the multiple stages of sensory systems is key for understanding how higher order representations are computed.

Over decades, feature selectivity has been studied based on tuning properties of individual neurons, and key transformations across stages of the sensory systems have been identified based on the presence or the absence of particular feature selectivity properties in each stage. The typical textbook example is contour orientation tuning and its emergence in the visual cortex, established by Hubel and Wiesel based on the observation of this type of tuning in the primary visual cortex and not in the thalamus [24,25]. Strikingly however, it is difficult to find a similar example of a simple feature emerging across a single synapse [26]. In the auditory system, multiple studies identified some tuning to various types of features at almost all stages after the cochlea. For example, tuning to the direction of frequency modulations over time has been described in octopus cells of the ventral cochlear nucleus [27], in the inferior colliculus [21,28] and then in the auditory thalamus [20] and cortex [29]. This ubiquity of simple feature tuning in single neurons across the auditory system could mean that specific feature representations emerge as soon as in the cochlear nucleus and are then propagated downstream. Alternatively, this could reflect a more continuous feature-extraction process, also suggested to some extent in the visual system in which some orientation and direction-selectivity is in fact observed as early as in the retina [30–32]. Distinguishing between these two alternatives is difficult because it requires precise comparisons of sound representations across fairly different brain areas based on the activity of noisy neurons exhibiting different types of tuning properties. The traditional approach is to choose a particular feature, such as sound modulation frequency or direction, quantify in single neurons the quality of tuning through simple indexes and compare across brain areas. Although intuitive, this approach has three limitations. First, tuning indexes often suppose a particular form of tuning which is not verified systematically in the data. For example, measuring the precision of frequency tuning is often performed by quantifying the width of the tuning curve around the preferred frequency [33–35]. This is operational for neurons that have a single preferred frequency but many neurons of the auditory system have multiple preferred frequencies [36]. Second, tuning indexes are easily corrupted by trial to trial variability that compounds both biologically relevant neural response variability and irrelevant measurement noise. For example, when measuring the preference of a single neuron for the direction of linear frequency sweeps, an index comparing responses to up- versus down-sweeps will capture more or less large differences solely due to response variability. At the single cell level, it is extremely

difficult to correct for this effect that leads to an overestimation of tuning strength which is hard to estimate. Third, single cell measurements do not provide a global measure of how well the collection of neurons in a given brain region encodes a particular feature. These shortcomings, and the fact that early stages of the auditory system are mostly sampled under anesthesia [37,38], have hindered so far the precise evaluation of feature selectivity progression across the auditory system.

We have shown recently that it is possible to measure the precision of auditory representations in a very generic manner with neural population metrics that are corrected for measurement noise and response variability [19]. Here we demonstrate that this approach can be used to systematically and generically measure tuning to any feature and even across features, independent of the form of single neuron tuning. We apply this method to a new dataset spanning three key stages of the auditory system, the cochlear nucleus, the inferior colliculus and the auditory cortex probed in unanesthetized mice with more than 300 sounds spanning simple and complex sound features. Our results show that feature tuning at the population scale systematically improves between from the periphery to the cortex partly gradually and partly abruptly with important steps at multiple levels of the auditory system. Tuning to some features is already fully established in the inferior colliculus, while for other, not necessarily more complex features, precise tuning only arises in the cortex.

## Results

### Large-scale recordings of neuronal activity in the unanesthetized, head-fixed mouse

To evaluate population tuning for a large set of relevant features in multiple regions of the mouse auditory system, we trained mice to stay head-fixed held by a headpost in a sound-proof box for the duration of the experiment (approximately 2 hrs, Fig 1A). In this setting, we performed large-scale recordings of single neurons in three regions of the auditory pathway: the cochlear nucleus (CN), the inferior colliculus (IC) and the auditory cortex (AC, Fig 1B) while the mouse was passively listening to a set of 307 sounds, mainly of 500 ms duration (between 50 and 500 ms), containing single frequency sounds either unmodulated (pure tones), amplitude-modulated (sinusoidally, AM, or linearly, Ramps), or frequency modulated (Chirps), multi-frequency sounds (sums of pure tones, Chords), white and filtered noise (WN), as well as complex natural-like sounds (Cpx), and decompositions of some of these complex sounds into smaller components to test for additivity of sound responses (Fig 1C). In the CN and IC, sound responses of neurons were recorded using Neuropixels 1.0 probes inserted with an angle predefined by stereotaxic coordinates during surgical head-post implantation, followed by spike-sorting using Kilosort 2.5 and manual curation of single units based on spike waveform and the presence of a 2 ms refractory period in the interspike interval distribution (Fig 1D, 1E). In the AC, neurons down to layer 5 were imaged using 2-photon calcium microscopy, and the calcium traces were linearly deconvolved [39], providing a temporal resolution of approximately 150 ms, which is sufficient to retrieve the temporal information necessary to discriminate 500 ms sounds and a few shorter sounds [19] (Fig 1F). After selecting units with a clear sound responsivity based on the reliability of their responses across multiple sound presentations (see "Materials and methods"), we retained 1,421 neurons from 21 experiments in 8 mice targeting the dorsal and ventral divisions of the cochlear nucleus, 551 neurons from 15 experiments in 9 mice targeting the central nucleus of the inferior colliculus (11 experiments from 7 mice targeting CN through IC with a slight bias towards more posterior regions of central IC and 4 experiments in 2 mice targeting only IC), and 4,217 neurons from 15 imaging sessions mainly in the primary areas of the auditory cortex AC (see "Materials and methods").

### General sharpening of sound representation tuning throughout the auditory system

Plotting responses of individual neurons to sample sounds shows a large response diversity across sounds and neurons in the three considered stages of the auditory system (Fig 2), reflecting the various neuronal subtypes and specializations present in these regions. Rich temporal variations of the firing rate are apparent mostly in electrophysiology recordings but also, at a lower resolution in calcium imaging recordings. This restricted temporal resolution is however not problematic

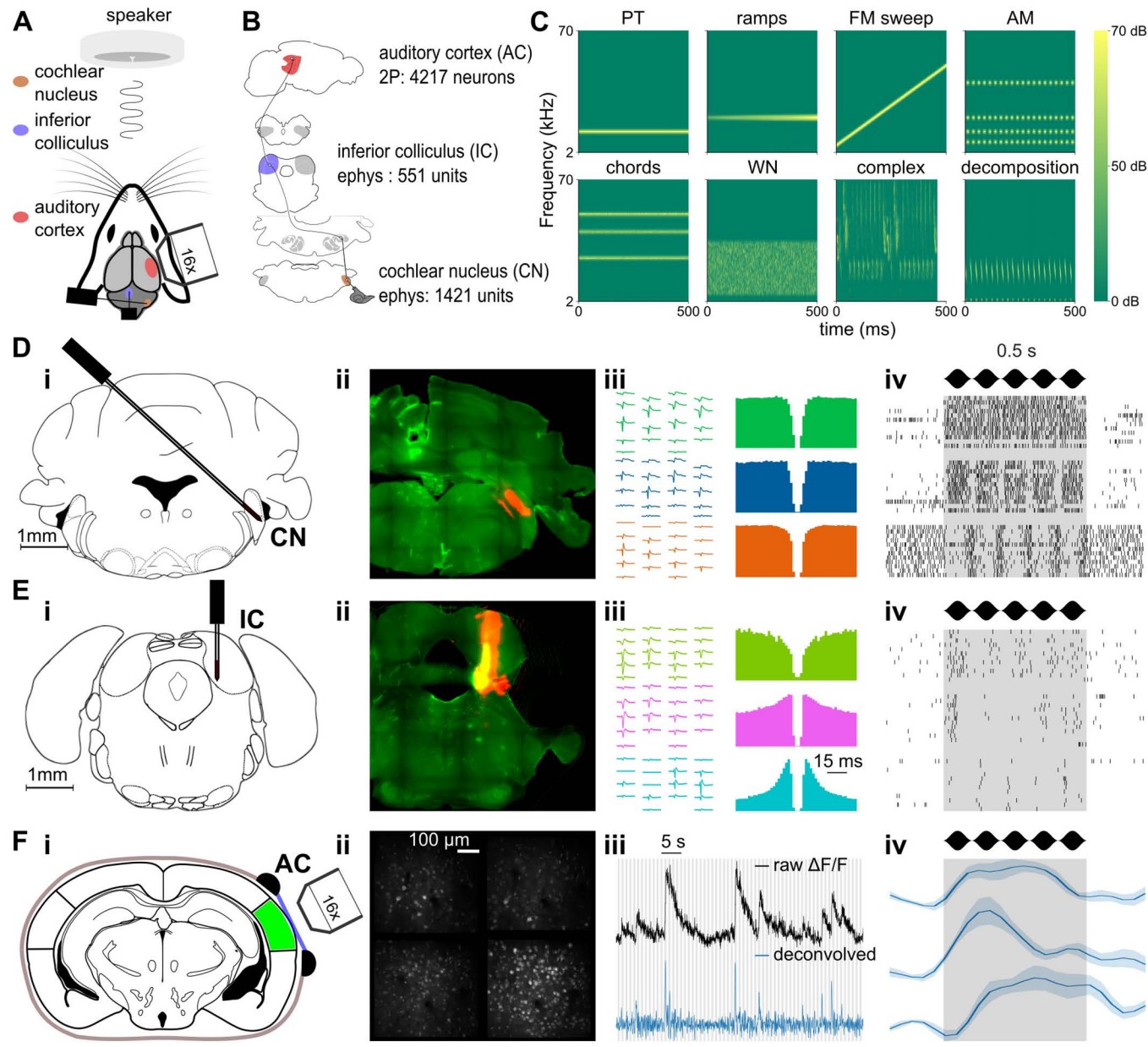

**Fig 1. Large-scale recordings of neuronal activity in head-fixed mice. A.** Schematic of the experimental setup. **B.** Schematic of the ascending auditory pathway. Each area where sound responses have been obtained via electrophysiology or 2-photon imaging is colored, with indication of the recording method and the number of single fibers/neurons. **C.** Sample spectrograms from each of the 8 categories contained in the 307 sounds stimulation set. **D-F.** Methodologies of data collection for each area. **D. i.** Schematic of the targeting of the cochlear nucleus using Neuropixels 1.0 probes **ii.** assessed via post-mortem histology. **iii.** Waveforms, auto-correlograms, **iv.** and raster plots of responses to a 10 Hz AM of 3 example units. **E.** Same as **D** for inferior colliculus dataset collection. **F. i.** Schematic of the imaging of the auditory cortex **ii.** and example image of 4 planes recording. **iii.** Raw ΔF/F trace (black) and deconvolved trace (blue) in one example neuron from the imaging session in **(i, ii)**. **iv.** Deconvolved response of 3 example neurons in response to a 10 Hz AM. The source data have been uploaded to https://doi.org/10.5281/zenodo.14421103.

because feature-specific tuning is defined as the specificity of the time-averaged firing rate of a neuron's response to particular stimuli, irrespective of response time course. Based on this definition, the representative examples shown in Fig 2 suggest that auditory cortex neurons have more specific responses, restricted to fewer sounds, as compared to

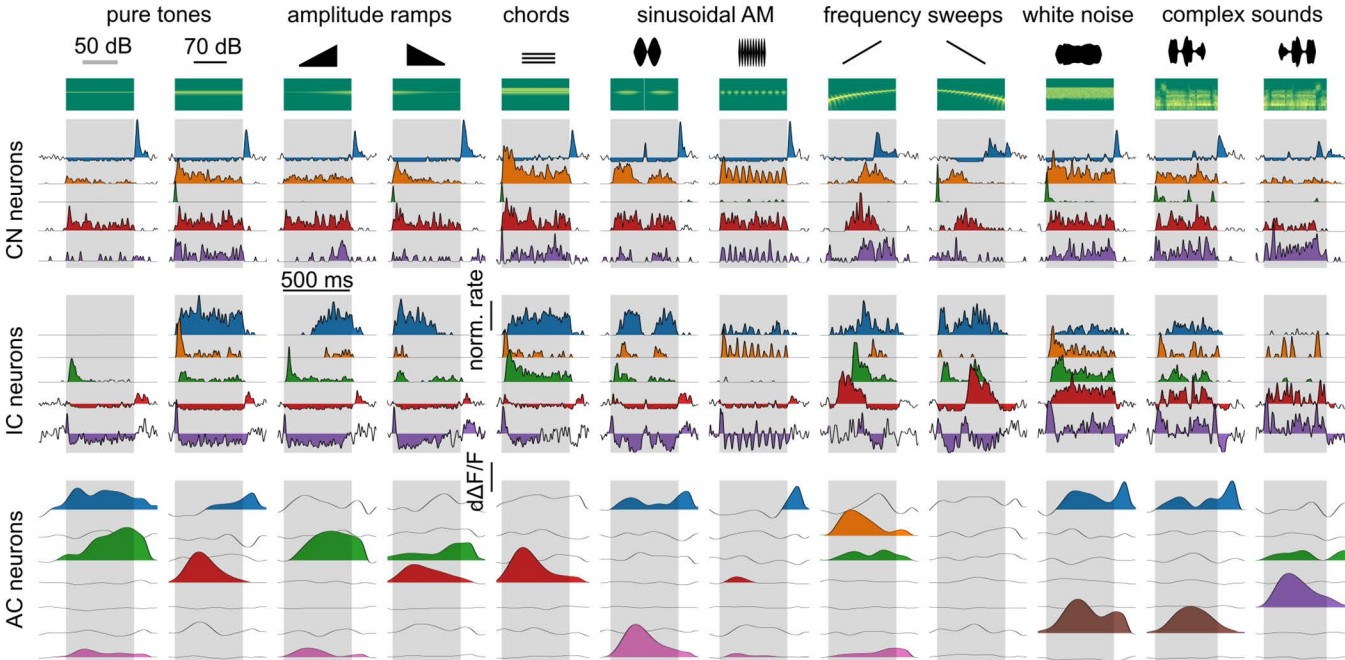

**Fig 2. Single cell sound response samples across the auditory system.** Trial-averaged responses of example neurons from cochlear nucleus (CN, 5 neurons), inferior colliculus (IC, 5 neurons), and auditory cortex (AC, 7 neurons) to 12 sounds with spectral content at 12 kHz (2 pure tones, 2 ramps, 1 chord, 2 AMs, 2 chirps, 1 WN and 2 complex, represented with their spectrograms). Sound presentation periods are shaded in gray. The source data have been uploaded to https://doi.org/10.5281/zenodo.14421103.

subcortical neurons. To quantify this, we reasoned that more specifically tuned neurons lead, at the population level, to patterns of activated neurons that are more dissimilar across sounds. Hence, the overall statistics of single neuron tunings can be summarized in a population tuning measure that globally assesses how similar neural representations are across sounds.

Concretely, if one considers a collection of N neurons with diverse tuning profiles for a range of pure tones (Fig 3A), each pure tone of frequency *f* produces a specific pattern of firing rates across the collection of neurons which is mathematically represented by a vector $V_f$ of dimension N (Fig 3B). We refer to this as the "spatial" representation of the tone since it depends on the space of all N neurons. Note that it is independent of their organization in anatomical space. The similarity between the patterns produced by two pure tone frequencies *f* and *f'* can be measured by a metric ρ ($\mathbf{V}_f$,$\mathbf{V}_{f'}$) (Fig 3B). To compare similarities across neuronal populations of different sizes it is necessary to use a normalized metric like the Pearson correlation coefficient or the cosine distance. Based on this metric (we used the Pearson correlation in this study), one can construct the matrix of similarities across representations of all tone frequency pairs (Fig 3B), previously termed representation similarity analysis (RSA) matrix [19,40]. The RSA matrix captures in a very generic manner the tuning of the neural population to pure tone frequencies. A line of the matrix corresponds to the tuning curve of the population to a particular frequency *f* (Fig 3C) i.e., how similar is the population pattern produced by frequency *f* to the patterns produced by all other frequencies. Likewise, by averaging similarity measures for all pairs of frequencies with the same frequency ratio, one can derive a population tuning curve for logarithmically spaced frequency intervals (Fig 3C), summarizing the overall precision of frequency tuning.

There are three main advantages of population tuning measurements with respect to averaging many single neuron tuning measures. First, they can be defined to reduce sensitivity to measurement noise and neural variability. As shown in

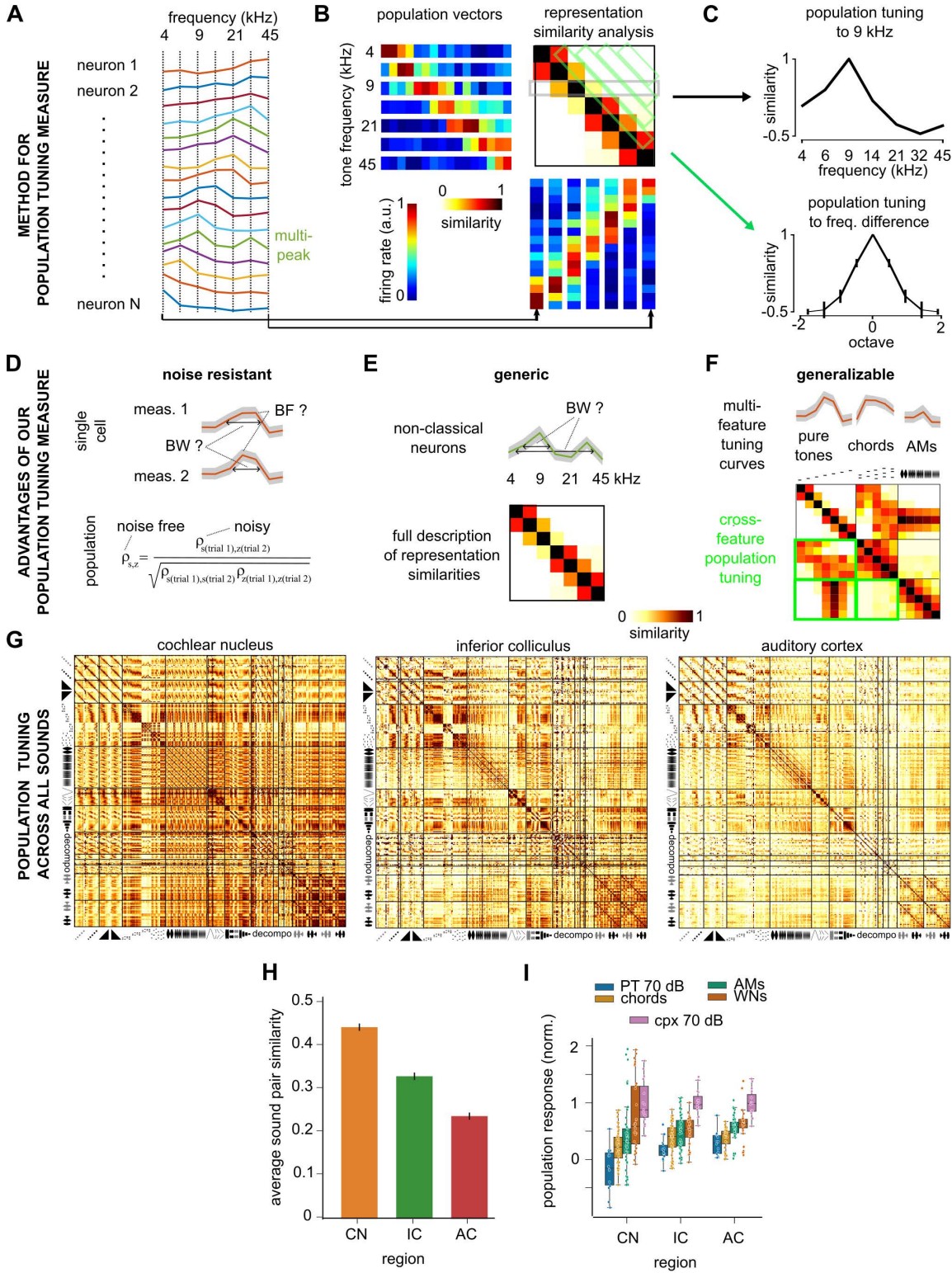

**Fig 3. Decorrelation of sound representations across the auditory system identified with a generic population tuning measure. A.** Synthetic tuning curves representing the time-averaged firing rate responses of 16 modeled neurons to seven pure tones. **B.** Construction of the population similarity analysis matrix displaying the similarity of the population representations of the seven pure tones for the synthetic measures shown in **A**. **C.** For

the same synthetic data, population tuning curves to 9 kHz (top) and for different frequency ratios (measured in octaves) as extracted from the population similarity matrix in **B**. **D–F**. Illustration of three advantages of population tuning measures against single cell measurements. **D**. While single cell measurements of tuning properties such as best frequency (BF) and frequency tuning band-width (BW) are imprecise and biased because of response variability, the effect of noise on population tuning measures can be corrected. **E**. While single cell tuning measures often depend on a response model (e.g., single frequency peak), population tuning provides a generic description of the similarity relationships between all sounds (e.g., all pure tone frequencies). **F**. Population tuning is a unified measure that generalizes across any sound feature or any sound. **G**. Matrices of spatial representation similarity for all regions. **H**. Average spatial representation similarity between all pairs of sounds computed from **G** for all regions. (Mean ± SEM: CN = 0.53 ± 1e − 3, IC = 0.41 ± 1e − 3, AC = 0.21 ± 1e − 3. Two-sample Wilcoxon sign-rank test for paired distribution between pairs of sounds across regions: CN against IC, $p < 1e − 63$, IC against AC, $p < 1e − 63$). **I**. Average population response of neurons to distinct categories of sounds, after standardization of each neuron by its maximum response and of the population by the mean response to complex sounds at 70 dB SPL. CN, cochlear nucleus, IC, inferior colliculus, AC, auditory cortex. The source data have been uploaded to https://doi.org/10.5281/zenodo.14421103.

previous studies [19,41], it is possible to build noise-corrected estimators of population metrics (Fig 3D). By contrast, on the single cell level, even a simple tuning measure like the width of tuning (e.g., distance at half-maximum) will be incorrectly estimated due to errors in the determination of the best frequency and the associated maximal response. These errors do not necessarily average out across a large number of neurons and are extremely difficult to correct. Second, the RSA matrix is a generic measure that describes all relationships between pairs of sounds. It therefore provides an evaluation of tuning that does not depend on a particular tuning model unlike, e.g., the single cell tuning width that requires a single peak tuning curve to be unambiguously defined (Fig 3E), or linear receptive field measures that are based on non-verified linearity assumptions [42]. Third, measuring the similarity of population patterns can be done meaningfully across any pair of sounds. Hence the same population tuning measure can be generalized to any type of feature and even applied across different types of features (e.g., pure tones frequency against amplitude modulation, Fig 3F).

Using our noise-corrected estimator of the Pearson correlation coefficient, we applied this generalized population tuning approach and computed RSA matrices for the representations of the 307 sounds in the cochlear nucleus, inferior colliculus and auditory cortex (Fig 3G). The first, obvious observation is that these plots show a progressive decrease of similarity between representations of different sounds from the cochlear nucleus to the cortex (Fig 3G, 3H). This indicates, as previously seen for a smaller and less rich set of sounds [19], that the patterns of neurons activated by different sounds become less and less overlapping or correlated as one progresses towards the auditory cortex. Using a detailed model of the auditory nerve [19], we evaluated the RSA matrix of the cochlear output and found that it largely resembles the RSA matrix of the cochlear nucleus (S1 Fig). This suggests that the decorrelation of sound representations mostly starts after the cochlear nucleus. Interestingly, we observed that this decorrelation process occurs without modifying the relative level of average population activity elicited by the different classes of sounds (Fig 3I). This indicates that the neural population-scale salience of sound representations [43] is largely determined by cochlear inputs despite a profound reorganization of the tuning properties.

## Improvement of tuning to complex sounds but not to pure tones

To better understand the decorrelation of representations observed across many sounds, simple and complex, we first measured the evolution of pure tone frequency tuning from the cochlear nucleus to the auditory cortex based on the population approach presented in Fig 3A–3C. Quantifying fine tuning at 0.3 octave distance, we observed very little evolution of the precision of the code for small frequency differences from cochlear nucleus to cortex with average correlation values around 0.55–0.6 across all considered stages (Fig 4A–4B, S1 Table). The same observation was made if we considered temporal information in the evaluation of population response specificity (S2 Fig). These results emphasize that the fine-scale precision of sound frequency tuning is already established in the cochlear nucleus and is not improved throughout the auditory system. We also observed that the half-width of pure tone tuning curves in single neurons is similar across the cochlear nucleus, inferior colliculus and auditory cortex, with a broad distribution of tuning widths at all stages (S2 Fig). This observation corroborates the population tuning measure. Frequency tuning can therefore not explain the decorrelation of sound

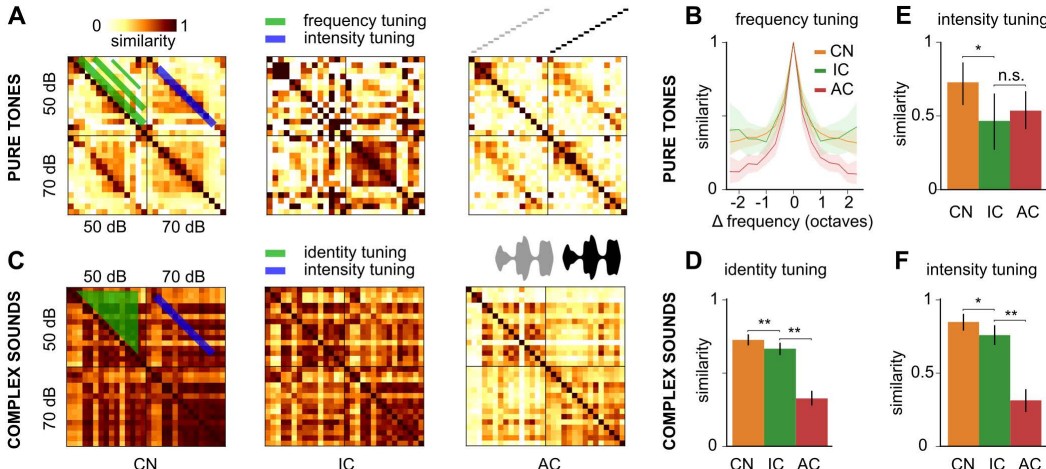

**Fig 4. Identity and intensity tuning improves differentially across the auditory system for simple and complex sounds. A.** Spatial representation similarity matrices for pure tones at 50- and 70-dB SPL. **B.** Evolution of spatial representation similarity between pure tones dependent on their frequency difference. **C.** Spatial representation similarity matrices for complex sounds at 50- and 70-dB SPL. **D.** Spatial representation similarity between different complex sounds with different identity (i.e., dolphin vs. bird call) at the same average intensity. **E.** Spatial representation similarity between pure tones at the same frequency but at different intensities. **F.** Spatial representation similarity between identical complex sounds at different intensity. CN, cochlear nucleus, IC, inferior colliculus, AC, auditory cortex. Full statistics are provided in S1 Table. The source data have been uploaded to https://doi.org/10.5281/zenodo.14421103.

representations observed across our diverse set of sounds (Fig 3H). By contrast, when we measured population tuning to the 15 complex sounds presented to the animals, we observed a small improvement from cochlear nucleus to inferior colliculus, and a large improvement in the auditory cortex (Fig 4C–4D). Hence, the auditory cortex decorrelates complex sound representations without improving pure tone tuning. Note that representation of larger frequency differences (>0.5) are more decorrelated in the cortex (Fig 4A–4B) which could contribute, in part, to the improvement of complex sound tuning. As our complex sounds typically contain a wide range of acoustic features beyond pure frequencies (Figs 1C, S2), this result suggests that the decorrelation of complex sound representations is based on the decorrelation of these features.

## Intensity tuning improves differentially for simple and complex sounds

We therefore decided to systematically evaluate how the tuning of each specific feature in our set of sounds evolves throughout the auditory system and started with the intensity feature. Using our population tuning approach, we measured the similarity of representations for the same pure tone or same complex sound but at two different intensities (50 and 70 dB SPL, Fig 4A, 4C). We observed that, for pure tones, intensity tuning is already present in the cochlear nucleus, further improves in the inferior colliculus, and stabilizes in the auditory cortex (Fig 4E, S1 Table). Strikingly, for complex sounds, intensity tuning evolves in a different manner, improving only slightly from cochlear nucleus to colliculus and markedly from colliculus to cortex (Fig 4F, S1 Table). Considering that complex sound identity tuning is also only fully established in the cortex, one could hypothesize that this striking result reflects the necessity of establishing identity tuning before intensity tuning.

## Early emergence of amplitude modulation tuning

We then focused on amplitude modulation which is a key feature for the recognition of auditory objects, e.g., in human speech [17,44]. At least two parameters should be considered when describing amplitude modulations: frequency, which relates to modulation speed, and direction (up versus down) [45,46]. We first quantified population tuning to modulation frequencies, using pure and multifrequency carrier sounds (8 and 12 kHz, 5 Chords, and white noise) (Fig 5A). Focusing

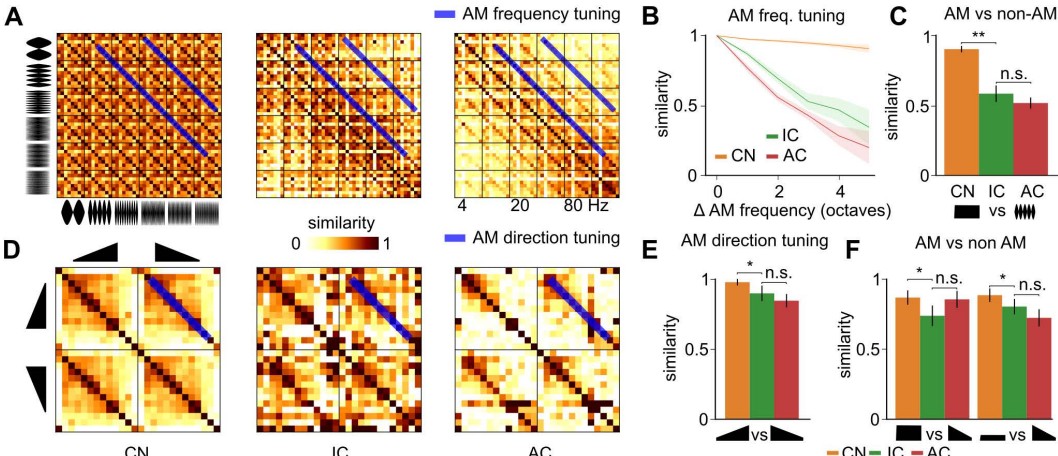

**Fig 5. Early emergence of amplitude modulation tuning. A.** Spatial representation similarity matrices for amplitude modulated (AM) sounds at 8 different carrier signals (pure tones and chords) and 6 modulation frequencies. **B.** Evolution of spatial similarity between AM sounds dependent on their modulation frequency difference. **C.** Similarity of the spatial representations of AM sounds and of the summed spatial representations of the pure tones corresponding to the carrier signal. **D.** Spatial representation similarity matrices for upward and downward linear intensity ramps (carrier signal = pure tone). **E.** Spatial representation similarity between upward and downward ramps at the same frequency. **F.** Spatial representation similarity between ramps and pure tones that have the same frequency and same start (*left*) or end (*right*) intensity as the ramp. CN, cochlear nucleus, IC, inferior colliculus, AC, auditory cortex. Full statistics are provided in S2 Table. The source data have been uploaded to https://doi.org/10.5281/zenodo.14421103.

on the 4–160 Hz modulation frequency range, we observed that tuning to amplitude modulations is poor in the cochlear nucleus, dramatically improves in the inferior colliculus and further improves, but only slightly, in the auditory cortex (Fig 5B, 5C, S2 Table). This result is in line with multiple previous reports showing a transformation from a pure temporal code in CN, exhibiting neurons phase-locked to an AM sound up to several hundreds of Hertz irrelevant of the modulation frequency [47,48], to a spatial code in IC, featuring band-pass, band-reject, and low-pass neurons [49]. When we included temporal information to compare population activity patterns, we observed a similar evolution of amplitude modulation tuning from cochlear nucleus to cortex (S3 Fig) indicating that neuron-specific preference for particular amplitude modulation frequency plays a major role in the representation of amplitude modulation frequencies already in the inferior colliculus. We also compared the population responses to AMs and the corresponding sum of pure tones of 500 ms duration (Fig 5C). To evaluate population tuning to amplitude modulation phase, we used linear pure tone intensity ramps of 500 ms oriented upwards or downwards. By contrast to AM frequency tuning, similarities were close to 1 in the cochlear nucleus (Fig 5E) indicating that, at this stage, AM direction is almost exclusively coded by the time course of neural responses (S3 Fig). Then population tuning for AM direction significantly improved in the inferior colliculus and further improved although not significantly in the auditory cortex (Figs 5E, S3, S2 Table). In a previous study with a larger neuron sample and multifrequency waveforms, a similar decrease from IC to AC was observed and was significant [19]. In the auditory cortex, the coding of intensity ramps is based on neurons that are specific to sound onset or offset but also to the sound intensity at offset and onset [39]. Therefore, tuning to linear ramps relates to the progressive establishment of pure tone intensity tuning (Fig 4E). In line with this process, the population representation of intensity ramps in the inferior colliculus became dissimilar to representations of the pure tones with the same frequency and the same onset or offset intensity (Fig 5F).

## Cortical improvement of tuning to chords versus pure tones

Natural sounds contain various frequency patterns which participate in the identification of the sound. We first studied harmonic and inharmonic combinations of pure frequencies that have a periodic or pseudo-periodic waveform. Our soundset includes a large number of frequency chords covering either low, middle, or high frequency ranges or the full frequency

range, as well as a few harmonic chords (Fig 6A). Tuning across different chords within the same frequency range (i.e., different combinations of the same subset of pure tones) was relatively stable across the auditory system with a small decrease in the inferior colliculus (Fig 6B left, S3 Table). Hence we did not observe a refinement in the representations of fine details in tone chords which is already accurate in the periphery, consistent with the lack of improvement of fine frequency tuning previously noted (Fig 4B). Tone chord representations were slightly dissimilar to the summed representations of the corresponding pure tones in the cochlear nucleus. This similarity level remained stable in the colliculus and decreased significantly in the auditory cortex (Fig 6C, S3 Table). Hence, the main transformation of chord representations in the auditory system is their differentiation from pure tones in the cortex. Taking in account temporal information in the similarity analysis did not alter this conclusion (S4 Fig).

## Progressive refinement of tuning to broad- and narrow-band noises

We next focused on broad-band frequency patterns which correspond to aperiodic, 'noise-like' sounds. To start exploring the tuning to broad-band sound parameters, we first quantified the similarity between population representations of filtered

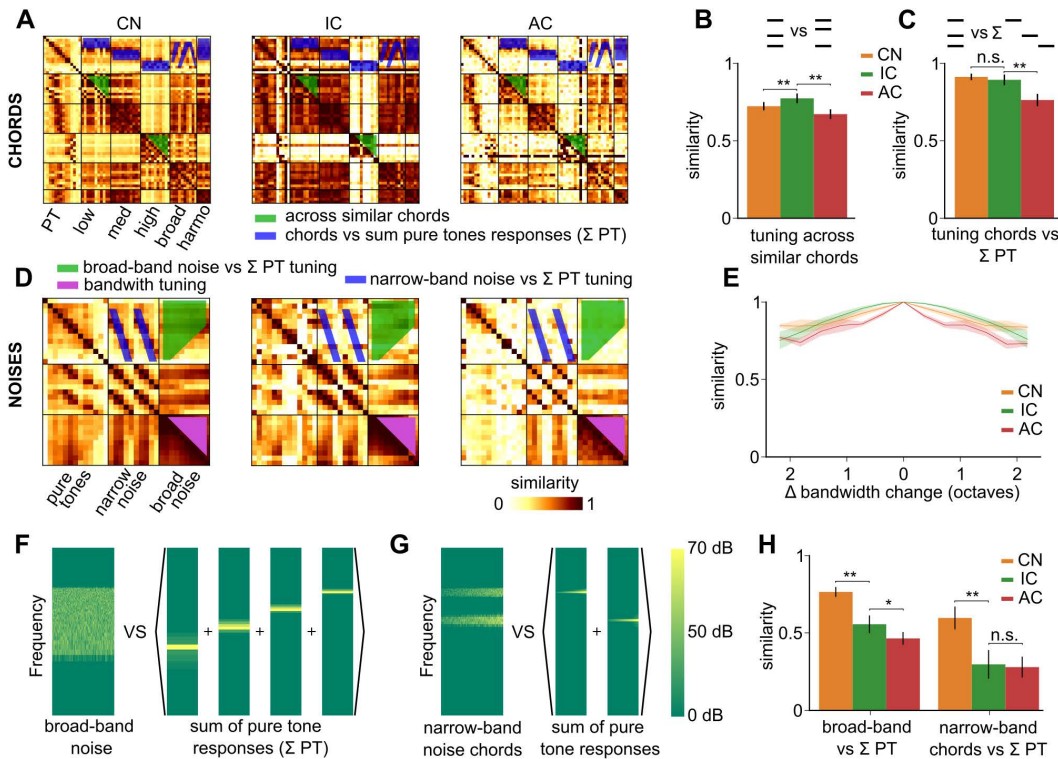

**Fig 6. Decorrelation of broad-band and multi-frequency sounds. A.** Spatial representation similarity matrices of pure tones at 70 dB SPL and their combination into various chords, organized based on the frequency range covered (low, mid, high, or full range) and on whether they are harmonic (harmo). **B.** Spatial representation similarity between chords built from the same pool of pure tones (*i.e.,* between all pairs of chords built from low, mid, high frequency pure tones but not between a low frequency chord and a mid frequency chord). **C.** Similarity between spatial representations of chords and the summed representations of the pure tones that compose them. **D.** Spatial representation similarity matrices of pure tones, narrow- and broad-band noises. **E.** Mean spatial representation similarity between broadband noises dependent on their difference in bandwidth across recorded areas. **F.** Schematic of the reconstruction of a 4.8-28 kHz broadband noise using spectrograms of sounds used **G.** Schematic of the reconstruction of a 12 + 25 kHz narrowband noise using spectrograms of sounds used. **H.** Similarity of spatial representations between broadband (left) or narrowband (right) noises and the summed response of pure tones included in their frequency range. CN, cochlear nucleus, IC, inferior colliculus, AC, auditory cortex. Full statistics are provided in S3 Table. The source data have been uploaded to https://doi.org/10.5281/zenodo.14421103.

noises across different bandwidths, ranging from 3 kHz to 80 kHz, logarithmically centered around 10 kHz (Fig 6D). Strikingly, we observed a weak tuning to noise bandwidth in all three considered brain regions (Fig 6E). A slight improvement of tuning was observed from inferior colliculus to cortex. Broadband noise representations also differed from the summed representations of all sampled pure tones within their bandwidth, and more so in the inferior colliculus and cortex with the largest improvement occurring between cochlear nucleus and inferior colliculus (Fig 6F–6H, S3 Table). Hence, like for amplitude modulations, most of the tuning to broadband sounds against pure tones is established in the inferior colliculus. Taking in account temporal information in the population tuning analysis did not substantially alter these observations (S4 Fig). Interestingly, broadband noises differ from pure tones in that their waveforms display large amplitude modulations of the envelope. Hence, the parallel improvement of tuning to broadband noise against pure tones and tuning to amplitude modulations (Fig 5) suggests that part of the encoding of broadband noises in the colliculus is based on the detection amplitude modulations.

## Tuning to the direction of frequency modulations and of complex sounds improves strongly in the auditory cortex

We next investigated population tuning to the direction of frequency variations in time. We observed that directional frequency modulation tuning slightly but significantly improved from the cochlear nucleus to the inferior colliculus (Fig 7A, 7B, S4 Table), consistent with the presence of directionally tuned cells in these structures [27, 50, 51]. However, a much larger improvement was observed from the inferior colliculus to the cortex (Fig 7A, 7B, S4 Table) consistent with our previous observations [19]. Interestingly, the weak specificity to direction masks a marked reorganization of frequency modulation representations in the inferior colliculus. Indeed, measuring tuning to frequency chirps against the summed representations of the pure tone tones included in the chirp's frequency range, we observed that it is already maximally improved in the colliculus (Fig 7C, S4 Table). The same effect is observed when taking in account temporal information in

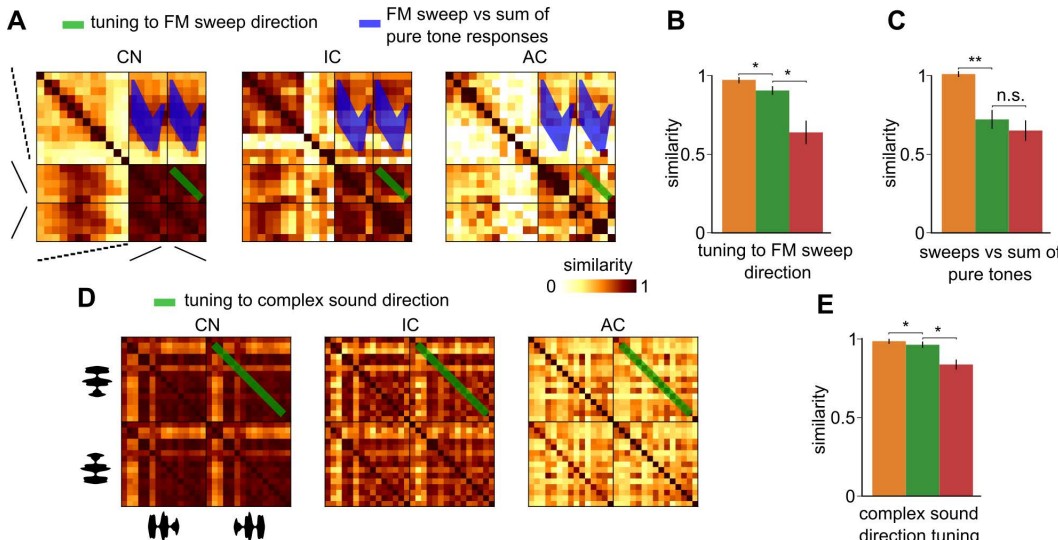

**Fig 7. Cortical decorrelation of sound direction. A.** Spatial representation similarity matrices of frequency modulated chirps at 70 dB SPL and pure tones at 70 dB SPL. **B.** Similarity of spatial representations between time-symmetric up- and down-frequency sweeps. **C.** Similarity of spatial representations of sweeps and of the summed spatial representations of pure tones traversed by the sweep. **D.** Spatial representation similarity matrices of forward and backward complex sounds at 70 dB SPL. **E.** Spatial representation similarity between forward and backward renditions of the same complex sound. CN, cochlear nucleus, IC, inferior colliculus, AC, auditory cortex. Full statistics are provided in S4 Table. The source data have been uploaded to https://doi.org/10.5281/zenodo.14421103.

our population tuning analysis (S5 Fig). Hence, like amplitude modulations, frequency modulations are processed in the inferior colliculus even if directionality tuning is strongly boosted in the auditory cortex.

We also measured the similarity between representations of complex sounds played forward and backward (Fig 7D) and observed that tuning to complex sound direction was poor in the cochlear nucleus and in the colliculus but strongly improved in the auditory cortex (Fig 7E, S4 Table). Hence, frequency modulations are processed subcortically but their direction, mostly encoded in the time course of responses before the cortex (S5 Fig), becomes efficiently represented by patterns of direction-specific neurons in the cortex [19], which may contribute to complex sound identity and direction tuning (Figs 4D and 7E).

Altogether, our results quantify the progressive transformation of sound representations from a spectrotemporal format at the periphery to a specific population tuning to features and combination of features in the cortex. Important transformations occur both in the colliculus and the cortex, in line with the recent observation that sound processing in colliculus is strongly non-linear [52]. This matches our observation that representation of complex sound tends to differ from the sum of the representations of their components (S5 Fig, S4 Table). Remarkably, our observations indicate that the global transformation of sound representations from the cochlear nucleus can be decomposed into gradual and stepwise changes for specific features, with some steps occurring in the brainstem and others afterwards (Fig 8).

## Discussion

This study introduces a noise-corrected population-scale approach to evaluate feature tuning in sensory systems and applies it to three key regions of the auditory system with a large set of sounds (Fig 1) comprising several features known to be relevant for sound recognition [46,53–56]. Thanks to this method and to an extensive neural recording dataset in unanesthetized mice, we describe the evolution of population-scale tuning to multiple features as summarized in Fig 8. Overall, our results demonstrate that tuning sharpens for diverse temporal and spectral features not coded by specific fibers of the auditory nerve. Importantly, our analysis focuses on the classical definition of response tuning, based on the time averaged response of the neurons. Adding temporal information improves population tuning, particularly in more peripheral regions. This reflects the fact that all the information necessary to identify acoustic features is already present at the auditory system input, for a large part under the form of temporal activity patterns. As shown previously, information

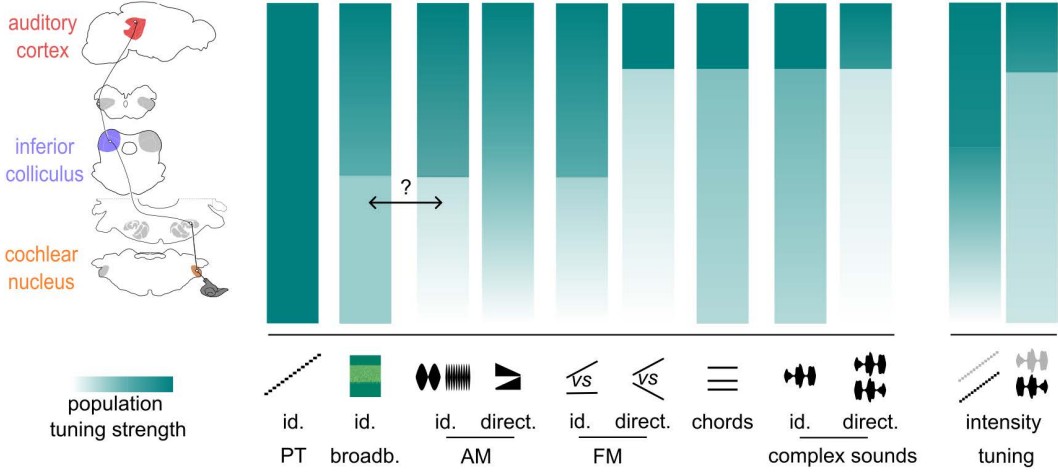

**Fig 8. Evolution of sound feature tuning along the auditory system.** Sketch representing population tuning strength across 4 different stages of the auditory (1 − similarity) for different acoustic features from pure frequency tones to amplitude (AM) or frequency (FM) modulations to complex sounds. The double arrow indicates a possible link between tuning to AM and to noises. id. = identity; direct. = direction; freq. = frequency.

present in the temporal domain progressively transfers to neuron-specific patterns along the auditory system [19]. This transformation is an essential computation of the auditory system which is specifically identified by tuning measures based on time-averaged activity.

From a global perspective, the refinement of population tuning we observe in the auditory system is analogous to the decorrelation processes observed, e.g., in the olfactory system [57,58] or in the hippocampal formation [59] which leads to a separation in the neural representation space of meaningful inputs. The decorrelation of representations has strong functional implications. Computational modeling indicates that decorrelated representations ease their memorization by neural networks [60]. We also showed previously that mice learn to discriminate faster sound pairs whose representations are more decorrelated [19,61]. Moreover, we previously observed in deep neural networks performing sound categorization (e.g., speech recognition) a similar decorrelation process that seems to be necessary for category formation [19]. Because it applies to both spectral and temporal features this decorrelation incorporates the concept of a temporal hierarchy in the auditory system with increasingly longer integration time windows from the cochlea to the cortex [62,63].

The idea of a decorrelation also relates to the information theoretic concept of redundancy reduction which was used previously to show in anesthetized cats that response specificity is higher in the cortex [64]. However, unlike these concepts, our approach permits us to specify the temporal features that are integrated in time, the features that become specifically encoded and where this happens in the auditory system. The refinement of population tuning also relates to the concept representation sparsening [65]. Loose patch recordings, a technique known to have little sampling biases, show that sound representations in the auditory cortex are based on sparsely responding neurons in the auditory cortex [66]. Response specificity to few sounds leads in single neurons to more specific population patterns and to decorrelation. Sparsely responding neurons, the majority of the neurons in the auditory cortex, also display low on-going firing rates [66,67]. This indicates that recording methods which, like multielectrode electrophysiology, preferentially sample highly firing neurons [68], may underestimate decorrelation. It however remains to identify which cell types underlie decorrelated representations in the auditory cortex and how they are sampled by different recording techniques. It also remains to precisely identify how feature tuning specifically evolves in secondary subfields of the auditory cortex, where they could refine in a task-specific fashion. High density sampling of these regions, much beyond the size of the present dataset, is necessary to solve this question.

Strikingly, however, tuning to frequency for small differences (0.3 octave) does not improve between the cochlear nucleus and the auditory cortex, as seen both at population and single cell levels (Figs 4 and S2), meaning that single frequency information is not further decorrelated, confirming a previous independent observation [19]. This result is surprising given the complex evolution of single cell pure tone tuning curves across the auditory system from V-shaped curves in the auditory nerve to a mixture of simple, multipeak and complex tuning curves in the cortex [36], and the existence of ultrasharp tuning curves in the auditory cortex [34]. Our results do not contradict these observations as, for example, the existence of few cells with ultrasharp pure tone tuning (not directly observable in single cells in our study due to the low resolution of our pure tone sample) can easily be compensated at the population level by a few more broadly tuned neurons. Some studies indicate that frequency tuning is under the control of local networks and can be modulated by manipulating neuronal subpopulations [69,70]. Our population-scale results suggest that these network processes maintain population tuning to frequency around a given setpoint across different stages of the auditory system which could be optimal for its operating constraints, while specific representations of other features are computed. Frequency tuning strength may therefore be an invariant of the auditory system, similar to the population level neuronal salience across sound categories which seems to increase mainly with the spectrotemporal complexity of the sound in all regions of the auditory pathway, without any equalization in higher stages of the system (Fig 3I).

Our systematic approach allowed us to identify features whose representations become decorrelated from each other, improving population tuning. This happens for temporal features including amplitude (Fig 5) and frequency (Fig 7) modulations, and for spectral features including broadband and multifrequency patterns, although the latter are mainly differentiated from pure tones and not between each other (Fig 6). The earliest improvement is observed for amplitude modulation

frequency or speed which mostly occurs in inferior colliculus (Figs 5 and 8), while, in parallel, frequency modulations differentiate from pure tone, intensity tuning starts establishing with the direction of intensity variations. By contrast, tuning to frequency modulation direction, to chords and to complex sound identity, intensity and direction fully develops later, in the auditory cortex (Fig 8). Hence, complex frequency patterns tend to decorrelate later than simple temporal modulations, with the exception of broadband sounds although their specific encoding might relate to the fact that they contain rich amplitude modulations (Fig 8) [71]. This highlights, most strikingly in the case of intensity decorrelation for pure tone as opposed to complex sounds, that features are not extracted independently from each other but that the computations are interdependent. Note that in some cases the variations of population tuning are small, especially when they start departing from a correlation value of 1. This is due to the nonlinearity of the Pearson correlation coefficient. We have shown in another study [19], that a change of population level correlation between 0.9 and 0.75 can increase learning speed 3-fold in a model aiming to discriminate two neural representations to produce two distinct behavioral responses. Hence, some of the small variations observed may have an important impact on downstream processing.

This description excludes the auditory thalamus, an essential stage of the auditory system. We have shown in a previous study that in the mouse auditory system, thalamic representations are more correlated to each other than both representations in the auditory cortex and in the inferior colliculus [19]. Therefore we did not expect to observe any improvement of tuning in the thalamic network. The global reduction of population tuning may be a specificity of the mouse auditory system and this should be more systematically studied across animal species. More specifically, the contribution of each region of the system to the extraction of one specific feature can be dependent on the special needs of a particular species. One striking example is the specialization of neurons in all regions of the auditory system of bats [72] responsible for the specificity of a majority of neurons in the IC to downward frequency sweeps [73], a prominent feature of echolocation calls. In comparison, such specialization is reduced in species that do not rely on echolocation such as rats [74] or mice [75]. The population tuning measurements introduced in this study could be easily generalized beyond the present dataset, not only in other species but also in other sensory modalities. They could also be applied to subdivisions of each of the structures studied, which was not possible here due to the sparse sampling of subfields. For example, the dorsal subdivision of the cochlear nucleus is known to show highly non-linear responses [76] and could implement more decorrelated representations than the ventral part. The generality of this approach could be helpful to reconcile the somewhat opposing views emerging from early works [15,25] in the visual system according to which certain feature (e.g., orientation selectivity) emerge in a specific stage of a sensory system, with modern observations that feature specificity can also occur earlier [30,31], an observation made decades ago in the auditory system. Our observations indicate that at the population scale tuning to a particular feature can arise early while staying globally weak, which is compatible with a restriction of tuning to a few neurons. However, we also observe that population tuning can evolve slowly but significantly and then be massively improved in a stepwise manner at a particular stage of the system (e.g., frequency modulation direction, Fig 7D). A similar phenomenon may occur for a wide range of features across sensory systems.

Understanding the mechanisms and role of the emergence of feature tuning in sensory systems has been an active field of research for several decades. Observations from artificial neural networks indicate that performance in object and category recognition is based on intermediate feature representations similar to those observed in the brain [19,77,78]. Moreover the progression of representations in cortical regions has analogies with the progression of representations in deep layers of convolutional neural networks [77,79,80]. However, to date there is no clear observation of stepwise progression of feature tuning in convolutional networks as we observed for certain features in the auditory system. A pioneering study [81] suggests that the compactness of the processing hierarchy influences the key processing steps which could explain species-related differences in the processing stage at which tuning to particular feature emergence in the visual system, but this type of hypothesis is not backed up by systematic, harmonized measurements as established in the present study. Therefore, our results provide precise indications for further improvement of models that aim at reproducing the transformation of auditory representations across stages of the auditory system.

 

## Materials and methods

### Ethics statement

All mice used for electrophysiology were 10–16 weeks old male C57Bl6J mice (26−30 g) that had not undergone any other procedures. Mice were group-housed (2–4 per cage) before and after surgery, had *ad libitum* access to food and water and enrichment (running wheel, cotton bedding and wooden logs) and were maintained on a 12-hr light–dark cycle in controlled humidity and temperature conditions (21−23°C, 45%–55% humidity). All experiments were performed during the light phase. All experimental and surgical procedures were carried out in accordance with the French Ethical Committees #89 (authorization APAFIS#27040-2020090316536717 v1).

### Surgical procedures

For electrophysiological recordings, we performed an initial surgery during which we exposed the bone above the dorsal cerebellum and positioned a head-post for reliable stereotaxic placement of the mouse head in the electrophysiology recording apparatus. Mice were injected with buprenorphine (Vétergesic, 0.05−0.1 mg/kg) 45 min prior to surgery. Induction of anesthesia was carried out using 3% isoflurane. After induction, mice were kept on a thermal blanket with their eyes protected with Ocrygel (TVM Lab), and anesthesia was maintained with 1.5% isoflurane delivered via a mask. Lidocaine was injected under the skin 5 min prior to incision. The skull above the inferior colliculus and cerebellum was exposed for ulterior craniotomy. A well was formed around it using dental cement in order to retain saline solution during recordings and the head post was fixed to the skull using cyanolit glue and dental cement (Ortho-Jet, Lang).

For calcium imaging, a craniotomy of 5 (AC) mm was performed above the AC. Injections of 150 nL of AAV1.Syn.GCaMP6s.WPRE (Vector Core, Philadelphia, PA; $10^{13}$ viral particles per ml; diluted 30×) were made at 30 nL/min with pulled glass pipettes at a depth of 500 μm and spaced every 500 μm to cover the large surface of the AC. The craniotomy was sealed with a circular glass coverslip. The coverslip and head post were fixed to the skull using cyanolit glue and dental cement (Ortho-Jet, Lang).

To protect the skull, the well was filled with a waterproof silicone elastomer (Kwik-Cast, WPI) that could be removed prior to recording for both imaging and electrophysiology. After surgery, mice received a subcutaneous injection of 30% glucose and metacam (1 mg/kg) and subsequently housed for one week with metacam delivered via drinking water or dietgel (ClearH20). After recovery, mice were trained to remain head-fixed for five days before recording by keeping them head-restraint for 30 min on day 1 up to 2 hrs on day 5.

Following this training and for electrophysiology, a craniotomy and a durectomy were performed above the cerebellum and/or inferior colliculus in a brief surgery under isoflurane anesthesia 1.5%, depending on the target region.

### Electrophysiological recordings

After at least one night of recovery, the unanesthetized mouse was head-fixed and Neuropixels 1.0 probes (384 channels) were inserted through the cerebellum at a 38–40° angle in the sagittal plane, targeting the contralateral cochlear nucleus, vertically targeting the inferior colliculus, or both. Electrode angle and entry point were defined relative to the initial head-post placement (Fig 1Ei, 1Fi). For cochlear nucleus targeting, fine tuning of these targeting parameters was progressively obtained through repeated penetrations based on time-locked responses to sounds easily detectable during probe insertion. Recording was started 15–20 min after the electrode position was locked to allow the brain tissue to stabilize and minimize movements of neurons in the first part of the recording. For post-hoc histological verification of the electrode track using fluorescent dye, the electrodes were dipped in diI, diO or diD (Vybrant Multicolor Cell-Labeling Kit, Thermofisher) prior to insertion. Recordings were performed using warmed saline filling the cyanolit glue well and in contact with the reference electrode. After each recording the well was amply flushed and then refilled with Kwik-Cast. Data was sampled at 30 kHz using a NI-PXI chassis (National Instruments) and the SpikeGLX acquisition software. Recording could

be repeated up to 4 days in a row to perform stable extracellular recordings. Using this approach, we recorded extracellular neuronal activity in the cochlear nucleus (inferior colliculus, respectively) during 21 (15) recording sessions in 8 (9) mice. At least 12 recordings targeted the ventral cochlear nucleus (VCN) and 4 the dorsal cochlear nucleus (DCN), as defined based on post-hoc histology (Fig 1Eii, 1Fii).

## Two-photon calcium imaging

Imaging was performed using an acousto-optic two-photon microscope (Karthala System, Orsay, France) combined with a pulsed laser (MaiTai, Spectra-Physics, Santa Clara, CA) set at 920 nm. As scanning is operated by acousto-optic deflectors based on MHz-range sound wave this microscope does not produce any audible background sound. We used a 16× objective (N16XLWD-PF, Nikon) to acquire images at 22.9 Hz from four planes interleaved by 50 μm in a field of view of 478 × 478 μm. In each animal, the auditory cortex tonotopic map was determined using intrinsic imaging, by measuring the change in 780 nm light reflectance over the cranial window, during presentation of pure tones with different frequencies in the lightly anesthetized mice (isoflurane) as described previously [82]. We then identified the main subfields of the right mouse auditory cortex according to [19,83], using the following definitions A1: most posterior and tonotopic field (low to high frequency gradient, ventro-posterior to dorso-anterior), AAF (Anterior Auditory Field): most anterior tonotopic field (low to high frequency gradient: dorsal to ventral), DP (Dorsal Posterior): non tonotopic, dorsal to A1, SRAF (SupraRhinal Auditory Field): small ventral tonotopic field posterior to AAF (low to high frequency gradient: dorsal to ventral), VPAF (Ventro Posterior Auditory Field): non-tonopic field posterio-ventral to A1. Using blood vessels as landmarks, we localized the neurons recorded with two-photon calcium imaging in the subfield map. A total of 20% of the neurons could not be localized because of the poor quality of intrinsic signals. The recorded neurons were mainly in primary auditory fields (A1 and AAF): A1 2,686 neurons, AAF: 303 neurons, DP: 358 neurons, SRAF: 6 neurons, VPAF: 12 neurons, Non-localized: 852 neurons.

## Sound set and experimental protocol

Sounds were generated with Python (The Python Software Foundation, Wilmington, DE) and were delivered at 192 kHz with Matlab (The Mathworks, Natick, MA), using a NI-PCI-6221 card (National Instruments) driven by a custom protocol using the Matlab Data Acquisition toolbox and feeding an amplified free-field loudspeaker (SA1 and MF1-S, Tucker-Davis Technologies, Alachua, FL) positioned in front of the mouse, 10–15 cm from the mouse ear. Sound intensity was cosine-ramped over 10 ms at the onset and offset to avoid spectral splatter. The head fixed mouse was isolated from external noise sources by sound-proof boxes (custom-made by Decibel France, Miribel, France) providing 30 dB attenuation above 1 kHz. Sound pressure levels were computed as Root Mean Square and calibrated at the location of the mouse ear for each sound, to compensate for frequency specific attenuations which were all smaller than 10 dB within the 4–30 kHz range).

The set of sounds consisted of 307 short sounds (<500 ms, sketched in Fig 1C) each repeated 12 times and played in a random order with a 1 s interval between sound onsets in 123 blocks of 30 sounds (total duration approximately 80 min). Twenty-eight Pure tones: pure tones at 14 frequencies logarithmically spaced between 2 and 60 kHz at 50 dB SPL and 70 dB SPL. Twenty-six Ramps: linearly ramped sounds, increasing and decreasing in intensity, at the same frequencies as the 13 pure tones between 2 and 50 kHz between 50 dB SPL and 70 dB SPL. Forty-eight Chords: summation of 2–4 70 dB pure tones from low (2, 4, 5, 8 kHz, 11 sounds), medium (10, 12, 16, 20 kHz, 11 sounds), high (25, 32, 40, 50 kHz, 11 sounds) frequency groups, broadly sampled frequencies (6, 12, 20, 40 kHz, 10 sounds) and harmonically arranged frequencies (4, 8, 12, 16 kHz, 5 sounds). Twenty Chirps: up- and down- frequency sweeps of different durations between 25 ms and 400 ms at 6–12 kHz, 50 dB SPL (10 sounds) or different frequency content between 4 and 50 kHz at 50 dB SPL in 500 ms (10 sounds). Thirty colored noises (WN): broadband noises at 50 dB SPL, 70 dB SPL or up-/down-ramped in 100 ms and 500 ms (6 sounds), filtered noises at different bandwidths between 2 and 80 kHz (14

sounds), and summation of two 1 kHz bandwidth filtered noises up- and down-ramped between 50 dB SPL and 70 dB SPL, matching frequencies of a subset of chords (10 sounds). Forty-eight amplitude-modulated sounds (AM): sinusoidally amplitude modulated sounds at 6 different modulation frequencies between 4 Hz and 160 Hz and 8 carrier frequency contents (2 pure frequencies, 5 sums of frequencies matching chords and 1 broadband noise). Sixty Complex sounds: 15 complex sounds (recordings of accelerated music, animal calls, and natural environments) high-pass filtered at 2 kHz, played forward or time reversed, at 50 dB SPL and 70 dB SPL. Forty-seven Decomposition sounds: snippets extracted from 4 selected complex sounds (2 bird calls, 1 dolphin call, 1 natural environment), and their compositions which reconstruct their associated complex sound. For the natural environment, the components were 5 filtered noises (in the range 2–12 kHz) ramped in amplitude (30–50 dB to 50–70 dB). Bird calls components were a sequence of fast, broad frequency chirps (2 chirps for the first, 4 chirps for the second) followed by trains of identical fast chirps and filtered noise (2 trains of noise clicks and 1 train of chirps at 55 Hz for the first, 3 trains of noise clicks and 2 trains of chirps at 44 Hz for the second). The sequence was repeated twice for reconstruction (Fig 6G). For the dolphin call, components was a sequence containing 2 fast chirps followed by trains of filtered noise clicks starting at 8.1 kHz and the five first harmonics of 11.6 kHz and modulated down by a 0.7 octave over 200 ms. This sequence was repeated twice for reconstruction.

## Histology

In order to extract the brain for histology, mice were deeply anesthetized using a ketamine-medetomidine mixture and perfused intracardially with 4% buffered paraformaldehyde fixative. The brains were dissected and left in paraformaldehyde overnight and then sliced into hundred micrometer sections using a vibratome and mounted. Analysis of the fluorescence band diI, diO or diD allowed isolating up to 3 tracks per mouse for electrophysiological experiments.

## Data preprocessing

Motion artifacts, regions of interest selection, and the signal extraction were carried out with the standard pipeline of the Python-based version of Suite2p [84]. For each region of interest, the mean fluorescence signal $F(t)$ was extracted together with the local neuropil signal $F_{np}(t)$. Then 70% of the neuropil signal was subtracted from the neuron signal to limit neuropil contamination. Baseline fluorescence $F_0$ was calculated with a sliding window computing the third percentile of a Gaussian-filtered trace over the imaging blocks. Fluorescence variations were then computed as $f(t) = \Delta F/F = (F(t) − F_0)/F_0$. An estimate of firing rate variations $r(t)$ was then obtained by linear temporal deconvolution of $f(t)$: $r(t) = f'(t) + f(t)/\tau$, $f'(t)$ being the first derivative of $f(t)$ and $\tau = 2$ s, the estimated decay of the GCAMP6s fluorescent transients. This simple method efficiently corrects the strong discrepancy between fluorescence and firing rate time courses due to the slow decay of spike-triggered calcium transients. It does not correct for the rise time of GCAMP6s, leading to remnant low pass filtering of the firing rate estimate and a delay of approximately 100 ms between the firing rate peaks and the peaks of the deconvolved signal. Finally, response traces were smoothed with a Gaussian filter ($\sigma = 31$ ms). Single trial sound responses were extracted (0.2 s before up to 0.8 s after sound onset) and the average activity over the pre-stimulus period (0.2–0.02 s before sound onset) was subtracted for each trial.

For electrophysiology, raw data were band-pass filtered (300–6,000 Hz) and channels from the electrode tip (corresponding to the cochlear nucleus region) were selected using SpikeInterface (https://github.com/SpikeInterface). Isolated clusters were identified using Kilosort 2.5 followed by manual curation based on the interspike-interval histogram and the inspection of the spike waveform using Phy (https://github.com/cortex-lab/phy). Canonical spike sorting was first applied with common parameters throughout the whole recording, attempting to optimize the spike detection and assignment to clusters. The measure of drift throughout the recording computed with Kilosort 2.5 showed minimal slow drift throughout the recording, except for some experiments in the first 10 min. Due to the high spiking activity in the cochlear nucleus, optimal spike sorting was achieved using lower Kilosort 2.5 parameters (detection threshold = 6, clustering threshold = 6, and matching thresholds = [4, 9]) in this region than for the inferior colliculus (detection threshold = 8, clustering

threshold = 8, and matching thresholds = [4, 10]). After manual curation, single trial sound responses were extracted (0.3 s before up to 1 s after sound onset) as a histogram of 1ms time bin and the average activity over the prestimulus period (0.3–0 s before sound onset) was subtracted for each trial. Based on histology, we identified a number of units whose location on the Neuropixel probe was not compatible with a localization in the cochlear nucleus or inferior colliculus. Finally, we selected units with at least minimally reliable response to sounds, by computing the inter-trial correlation of the temporal response for each sound and keeping units with average correlation over all sounds above 0.05 (electrophysiology data, time bin = 5 ms) or 0.3 (2-photon data, time bin = 42 ms).

## Noise-corrected correlation

For each dataset, population representations were estimated after pooling all recording sessions in a virtual population and shuffling trial identity for each sound to limit noise correlations between neurons. We used the correlation between population vectors as a metric of similarity between representations. The areas and techniques used to estimate neuronal ensemble representations yielded different levels of trial-to-trial variability due to intrinsic neuronal response variability and measurement noise. Most representation metrics are biased by variability, even after trial averaging, due to variability residues. Here, we use a noise-corrected Pearson correlation metric: the value of the Pearson correlation coefficient $\rho_{\vec{\nu}_s \, \vec{\nu}_{s'}}$ between population vectors for two sounds $\vec{\nu}_s$ and $\vec{\nu}_{s'}$ in absence of variability can be exactly estimated from noise-corrupted single-trial observations $\vec{\nu}_{s,r}$ and $\vec{\nu}_{s',r}$ of $\vec{\nu}_s$ and $\vec{\nu}_{s'}$ when their dimension N approaches infinity, based on the formula:

$$\rho_{\vec{\nu}_s \, \vec{\nu}_{s'}} \approx \frac{\frac{1}{R^2} \sum_{r,r'} \rho_{\vec{\nu}_{s,r} \, \vec{\nu}_{s',r'}}}{\sqrt{\frac{1}{R^2(1-R)^2} \left( \sum_{r \neq r'} \rho_{\vec{\nu}_{s,r} \, \vec{\nu}_{s,r'}} \right) \left( \sum_{r \neq r'} \rho_{\vec{\nu}_{s',r} \, \vec{\nu}_{s',r'}} \right)}}$$

in which r and r' are single trial indices and R is the total number of trials [19].

This estimator can yield values that can be outside [−1, 1] when $\frac{1}{R(1-R)} \sum_{r \neq r'} \rho_{\vec{\nu}_{s,r} \, \vec{\nu}_{s,r'}}$ approaches 0, i.e., for representations that are dominated by noise. Thus, we clipped values to remain in the range [−1, 1]. As significant noise-correlations in single-trials can lead to an effective N below the actual number of neurons, we defined r and r' as the average response over the two halves of trials randomly sampled 20 times.

To evaluate the significance of mean correlation differences across all sound pairs for temporal and rate representations, we used a two-sample Wilcoxon signed-rank test.

Spatio-temporal correlation was measured on vectors formed by concatenating the responses of all neurons throughout time (vector dimension = $N_{\text{Neurons}} \times N_{\text{TimeBins}}$). Spatial correlation was measured first by time-averaging the responses of each neuron and then concatenating these values for all neurons (vector dimension = $N_{\text{Neurons}}$). In both cases, we used data from the sound onset to 100 ms after the sound offset. The time bin of spatio-temporal vectors is 5 ms for electrophysiology data (CC, CN, IC) and 43.5 ms for calcium imaging (AC).

## Auditory nerve model

The auditory nerve model is an adaptation of the seminal model of Meddis [85,86] to the mouse cochlea, validated with CBA mouse auditory nerve recordings during pure tone recordings [87]. The model consists of a cascade of six stages recapitulating stapes velocity, basilar membrane velocity, inner hair cell (IHC) receptor potential, IHC presynaptic calcium currents, transmitter release events at the ribbon synapse, and the probability of spike generation in auditory nerve fibers (ANFs) including refractory effects. The input to the model is a sound waveform (in Pascals). The output is the probability of spiking events in 691 ANFs innervating 40 IHCs with a characteristic frequency (CF) distributed at regular intervals along the cochlear tonotopic from 5 to 50 kHz, 12 IHCs per octave. This distribution covered 82.8% of the basilar

membrane length from 1.2% (apex) to 83.9% (base) in 2.07% increments. According to experimental data, the number of ANFs per IHC ($N$) was controlled by the relationship $N = -0.0038x^2 + 0.375x + 7.9$ where $x$ is the IHC location along the basilar membrane such that $x = -56.5 + 82.5 \log_{10}(CF)$, with $x$ in percent from the apex and $CF$ in kHz. By adjusting the time constant of the calcium clearance $\tau_{Ca}$ within each IHC synapse, ANFs with different spontaneous discharge rate ($SR = 91.1 \, \tau_{Ca}^{2.66}$, with $\tau_{Ca}$ in ms and $SR$ in spikes/s) were simulated from 0.5 to 95 spikes/s ($21 \pm 19.8$ spikes/s, mean $\pm$ SD) to match the $SR$ distribution reported in mouse auditory nerve.

## Supporting information

**S1 Fig.** Comparison of representations between cochlear nucleus and auditory nerve fiber model. **A.** Sounds sampled at 400 kHz (represented here as a spectrogram) are fed into a detailed biophysical model of the cochlea and auditory nerve. The model simulates passive basilar membrane properties (1), stereocilia transduction (2, 3), calcium channel dynamics in hair cells (4) and synaptic release, as summarized in the "Materials and methods" section. **B.** Responses of example neurons from the auditory nerve fiber model with spectral content at 12 kHz (2 pure tones, 2 ramps, 1 chord, 2 AMs, 2 chirps, 1 WN and 2 complex, represented with their spectrograms). Sound presentation periods are shaded in gray. **C.** Matrices of spatial representation similarity for the auditory nerve fiber model activity during sound presentation and for cochlear nucleus data as in Fig 3. **D.** Average spatial representation similarity between all pairs of sounds computed from **C** for the auditory nerve fiber model and for cochlear nucleus data. The source data have been uploaded to https://doi.org/10.5281/zenodo.14421103.
(TIFF)

**S2 Fig.** Representation similarity of sound identity and intensity when including temporal information. **A and B.** Sample spectrograms of the sounds compared in the population tuning analyses presented in Figs 4 and S2C–S2H separating identity and intensity comparisons. **C–H** Same measures of population tuning as in Fig 4A–4F but taking in account temporal information by comparing the concatenated population vector time series describing the full spatio-temporal population representation over the duration of the sound instead of using only the time-averaged activity. Temporal information does not change representation similarity for pure tones and pure tone intensity (**C–E**), but decreases similarity for complex sound identity and intensity. **F.** Distributions of the halfwidth of pure tone tuning curves (70 dB) for all responsive neurons of each datasets. The distribution profiles are shown as violin plots. **G.** Same as **F.** displayed as overlapping histograms with 0.3 octave bins. (**H–J**). For identity, the result is expected as complex sounds include a large amount of temporal information. Complex sound decorrelation is therefore less pronounced for the spatio-temporal than for the time-averaged spatial representation (Fig 4E), in line with the observation that a key computation of the auditory cortex is to encode temporal information into specific spatial patterns [19]. The improvement of intensity tuning for complex sounds (**J**) and not for pure tones after including temporal information was also observed previously [19]. CN = cochlear nucleus, IC = inferior colliculus, AC = auditory cortex. Full statistics are provided in S5 Table. The source data have been uploaded to https://doi.org/10.5281/zenodo.14421103.
(TIFF)

**S3 Fig. Representation similarity of amplitude modulations when including temporal information. A.** Spectrogram of amplitude modulated sounds. **B.** Spectrogram summarizing the comparison between AM and non-AM activity. **C.** Spatio-temporal representation similarity matrices for 48 AM sounds at 8 carrier frequency contents and 6 modulation frequencies. **D.** Evolution of spatio-temporal similarity between AMs dependent on their modulation frequency difference. **E.** Spatio-temporal representation similarity between AM sounds and the non-modulated carrier signal. **F.** Spatio-temporal representation similarity matrices for upward and downward linear intensity ramps. **G.** Spatio-temporal similarity between upward and downward Ramps at the same frequency. **H.** Similarity between intensity ramps and pure tones at the same

frequency and at the start (left) or end (right) intensity of the ramp. CN = cochlear nucleus, IC = inferior colliculus, AC = auditory cortex. Full statistics are provided in S6 Table. The source data have been uploaded to https://doi.org/10.5281/zenodo.14421103.
(TIFF)

**S4 Fig.** Representation similarity of broad-band and multi-frequency sounds when including temporal information. **A.** Spectrogram of different chords generated from the same pool of pure tones and of noises with different bandwidths. **B.** Spectrogram sketching the comparison between representation of 3 chords and the summed representations of the corresponding pure tones (top). Spectrogram sketching the comparison between representations of 4 noises and the summed representations of the pure tones included in their frequency range (bottom). **C.** Spatio-temporal representation similarity matrices of pure tones at 70 dB SPL and their summation into chords, organized based on their frequency content. **D.** Similarity between chords built from the same pool of pure tones. **E.** Similarity between spatio-temporal representations of chords and the summed representations of the pure tones that compose them. **F.** Spatio-temporal representation similarity matrices of pure tones, narrow- and broad-band noises. **G.** Spatio-temporal representation similarity between broadband noises dependent on their difference in bandwidth across recorded areas. **H.** Similarity of spatio-temporal representations between broadband (left) or narrowband (right) noises and the summed response of pure tones included in their frequency range. CN = cochlear nucleus, IC = inferior colliculus, AC = auditory cortex. Full statistics are provided in S7 Table. The source data have been uploaded to https://doi.org/10.5281/zenodo.14421103.
(TIFF)

**S5 Fig. Representation similarity for sound direction when including temporal information. A.** Representational similarity matrices of pure tones at 70 dB SPL and Chirps at 70 dB SPL that span the equivalent frequency range for the full spatio-temporal description of population activity in all regions. **B.** Similarity between same frequency up- and down-frequency modulated chirps for spatio-temporal population activity. **C.** Similarity between Chirps and the response reconstructed from pure tones traversed by the Chirp, for the spatial code. **D.** Representational similarity matrices of forward and backward Complex sounds at 70 dB SPL for spatio-temporal population activity in all regions. **E.** Similarity between the same complex sounds in both temporal directions for spatio-temporal population activity. CN = cochlear nucleus, IC = inferior colliculus, AC = auditory cortex. CN = cochlear nucleus, IC = inferior colliculus, AC = auditory cortex. Full statistics are provided in S8 Table. The source data have been uploaded to https://doi.org/10.5281/zenodo.14421103.
(TIFF)

**S6 Fig.** Specificity of complex sound representations against the sum of their decomposition. **A.** Schematic of the reconstruction process of sounds as a sum of individual components (top) and responses of 2 example neurons from the cochlear nucleus to the example complex sound, its components, its reconstruction, and its reconstructed response). **B.** Similarity between the responses to reconstructed complex sounds and reconstructed response from responses to components of the complex sound for 3 complex sounds for both codes. **C.** same as **B** but for the full spatio-temporal population activity. CN = cochlear nucleus, IC = inferior colliculus, AC = auditory cortex. Full statistics are provided in S8 Table. The source data have been uploaded to https://doi.org/10.5281/zenodo.14421103.
(TIFF)

**S1 Table. Frequency and intensity statistics for the spatial code.** Table summarizing the values and statistics of data plotted in Fig 4. For each row, the top value is Mean ± SEM for the region and the bottom value is the Wilcoxon rank-sum test between the region and the previous region (IC against CN, and AC against IC). Identity coding: Pure tones, $N = 8-3$ sound pairs for 0.3−2.1 octave difference; Complex, $N = 105$ sound pairs. Intensity coding: Pure tones, $N = 14$ sound pairs; Complex, $N = 15$ sound pairs. Significant differences are marked in bold.
(DOCX)

**S2 Table. Amplitude modulation statistics for the spatial code.** Table summarizing the values and statistics of data plotted in Fig 5. For each row, the top value is Mean ± SEM for the region and the bottom value is the Wilcoxon rank-sum test between the region and the previous region (IC against CN, and AC against IC). Significant differences are marked in bold. Periodic modulations $N=25$ sound pairs for each difference; Periodic modulations against pure tones, $N=42$ sound pairs; Linear modulations $N=13$ sound pairs; Linear modulations against pure tones, $N=13$ sound pairs.
(DOCX)

**S3 Table. Multi-frequency statistics for the spatial code.** Table summarizing the values and statistics of data plotted in Fig 6. For each row, the top value is Mean ± SEM for the region and the bottom value is the Wilcoxon rank-sum test between the region and the previous region (IC against CN, and AC against IC). Significant differences are marked in bold. Chords $N=195$ sound pairs; Chords against pure tones, $N=50$ sound pairs; Noise bandwidth, $N=8−3$ sound pairs for 0.5−3 octave difference; Noise again.
(DOCX)

**S4 Table. Frequency modulation statistics for the spatial code.** Table summarizing the values and statistics of data plotted in Fig 7. For each row, the top value is Mean ± SEM for the region and the bottom value is the Wilcoxon rank-sum test between the region and the previous region (IC against CN, and AC against IC). Significant differences are marked in bold. Sweep direction $N=5$ sound pairs; Sweep against pure tones, $N=10$ sound pairs; Complex direction, $N=15$ sound pairs.
(DOCX)

**S5 Table. Frequency and intensity coding statistics for the spatio-temporal code.** Table summarizing the values and statistics of data plotted in S2 Fig. For each row, the top value is Mean ± SEM for the region and the bottom value is the Wilcoxon rank-sum test between the region and the previous region (IC against CN, and AC against IC). Identity coding: Pure tones, $N=8−3$ sound pairs for 0.3−2.1 octave difference; Complex, $N=105$ sound pairs. Intensity coding: Pure tones, $N=14$ sound pairs; Complex, $N=15$ sound pairs. Significant differences are marked in bold.
(DOCX)

**S6 Table. Amplitude modulation statistics for the spatio-temporal code.** Table summarizing the values and statistics of data plotted in S3 Fig. For each row, the top value is Mean ± SEM for the region and the bottom value is the Wilcoxon rank-sum test between the region and the previous region (IC against CN, and AC against IC). Significant differences are marked in bold. Periodic modulations $N=25$ sound pairs for each difference; Periodic modulations against pure tones, $N=42$ sound pairs; Linear modulations $N=13$ sound pairs; Linear modulations against pure tones, $N=13$ sound pairs.
(DOCX)

**S7 Table. Multi-frequency statistics for the spatial code.** Table summarizing the values and statistics of data plotted in S4 Fig. For each row, the top value is Mean ± SEM for the region and the bottom value is the Wilcoxon rank-sum test between the region and the previous region (IC against CN, and AC against IC). Significant differences are marked in bold. Chords $N=195$ sound pairs; Chords against pure tones, $N=50$ sound pairs; Noise bandwidth, $N=8−3$ sound pairs for 0.5−3 octave difference; Noise against pure tones, $N=10$ sound pairs for Broad, 8 sound pairs for Ramps.
(DOCX)

**S8 Table. Frequency modulation statistics for the spatial code.** Table summarizing the values and statistics of data plotted in S5 and S6 Figs For each row, the top value is Mean ± SEM for the region and the bottom value is the Wilcoxon rank-sum test between the region and the previous region (IC against CN, and AC against IC). Significant differences are marked in bold. Sweep direction $N=5$ sound pairs; Sweep against pure tones, $N=10$ sound pairs; Complex direction, $N=15$ sound pairs; Reconstruction (spatial), $N=4$ sound pairs; reconstruction (spatio-temporal), $N=4$ sound pairs.
(DOCX)

## Acknowledgments

We thank Maia Brunstein of the Hearing Institute Bioimaging Core Facility of C2RT/C2RA for help in acquiring histology images. The authors did not use generative AI and AI-assisted technologies at any step of the study and in the writing process.

## Author contributions

**Conceptualization:** Etienne Gosselin, Brice Bathelier.

**Data curation:** Etienne Gosselin.

**Formal analysis:** Etienne Gosselin.

**Funding acquisition:** Sophie Bagur.

**Investigation:** Etienne Gosselin.

**Methodology:** Etienne Gosselin, Sophie Bagur, Sara Jamali, Jean-Luc Puel, Jérôme Bourien, Brice Bathelier.

**Project administration:** Brice Bathelier.

**Software:** Etienne Gosselin, Jean-Luc Puel, Jérôme Bourien.

**Supervision:** Sophie Bagur, Brice Bathelier.

**Validation:** Etienne Gosselin, Brice Bathelier.

**Writing – original draft:** Etienne Gosselin, Sophie Bagur, Brice Bathelier.

**Writing – review & editing:** Etienne Gosselin.

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
