## [Editor Report · Decision Letter 0]

25 Apr 2025

Dear Dr Bathelier,

Thank you for submitting your manuscript entitled "Feature-dependent decorrelation of sound representations across the auditory pathway" for consideration as a Research Article by PLOS Biology.

Your manuscript has now been evaluated by the PLOS Biology editorial staff as well as by an academic editor with relevant expertise and I am writing to let you know that we would like to send your submission out for external peer review.

Once your full submission is complete, your paper will undergo a series of checks in preparation for peer review. After your manuscript has passed the checks it will be sent out for review. To provide the metadata for your submission, please Login to Editorial Manager (https://www.editorialmanager.com/pbiology) within two working days, i.e. by Apr 27 2025 11:59PM.

Kind regards,

Christian

Christian Schnell, PhD

Senior Editor

PLOS Biology

cschnell@plos.org

---

## [Decision Letter · Decision Letter 1]

2 Jul 2025

Dear Dr Bathelier,

Thank you for your patience while your manuscript "Feature-dependent decorrelation of sound representations across the auditory pathway" was peer-reviewed at PLOS Biology. It has now been evaluated by the PLOS Biology editors, an Academic Editor with relevant expertise, and by several independent reviewers.

In light of the reviews, which you will find at the end of this email, we would like to invite you to revise the work to thoroughly address the reviewers' reports.

As you will see below, the reviewers have many positive comments about your study. The most important issue both reviewers raise is the temporal difference between the calcium imaging and the electrophysiological recordings. Addressing this issue may require additional experimental data to address this concern. Please also address the other reviewer comments in full.

Given the extent of revision needed, we cannot make a decision about publication until we have seen the revised manuscript and your response to the reviewers' comments. Your revised manuscript is likely to be sent for further evaluation by all or a subset of the reviewers.

**IMPORTANT - SUBMITTING YOUR REVISION**

*Re-submission Checklist*

*Published Peer Review*

*PLOS Data Policy*

*Blot and Gel Data Policy*

Sincerely,

Christian

Christian Schnell, PhD

Senior Editor

PLOS Biology

cschnell@plos.org

REVIEWS:

Reviewer #1 (Tetsufumi Ito signed his report): The manuscript by Gosselin and colleagues recorded many neurons in the cochlear nucleus (CN), inferior colliculus (IC), and auditory cortex (AC) and analyzed the similarity of responses to various sound stimuli using Pearson's correlation. They concluded that the neuronal population in the CN responded more similarly to various stimuli than higher centers. In the IC and AC, the similarity of population activity decreased, suggesting sparser coding in the higher centers. The degree of reduction in similarity within the hierarchy differed among stimulus types: AM coding mainly occurred in the IC, whereas tuning to harmonic sounds emerged in the AC. The method is clear, and the paper is well-written. I have several comments related to the methodology of the study.

To record population activity, calcium imaging was employed for the AC, whereas multichannel electrodes were used for other regions. Obviously, the temporal resolution of calcium imaging (~150 ms) is inferior to unit recording (on the order of sub-milliseconds), as the authors mentioned, indicating that AC activity reflects time-averaged firing within the temporal resolution window. The authors concluded that this was not an issue because, in their study, feature-specific activity was defined as the average firing rate to the stimuli. However, it should be noted that in lower centers, neurons rely more on temporal coding (i.e., phase-locking to the stimulus time course) rather than rate coding, and this temporal coding is reshaped into rate coding in higher centers. The authors should at least discuss the similarity of temporal coding activity in the lower and higher centers.

Taking into account the time resolution of the calcium imaging (150 ms), the duration of the stimuli (500 ms) seems slightly too short. Some stimuli even have shorter durations. The authors should provide a justification for employing short-duration stimuli.

The interval between stimuli is set at 1 second. This is too short for the AC and some neurons in the IC, since higher-order auditory centers have a ~2-second time integration window (Shymkiv et al., 2025), which allows for adaptation to the stimuli. The authors should analyze the degree of adaptation caused by repetitive stimulation.

The authors found that frequency tuning specificity is already established in the CN and does not improve in the higher hierarchical centers. As they pointed out, this is surprising because tuning sharpness and non-linearity typically increase during hierarchical processing. They should analyze the raw firing properties of individual neurons to identify the factors that make the population's frequency tuning stable across the hierarchy. For other stimuli, it would be more informative to analyze the raw firing activity of each neuron and the relationships between firing activities while preserving the temporal information of the firing activities, which is lacking in the study's analysis.

Reviewer #2: This study aims to systematically study how sound feature representations evolve across the mouse auditory pathway from the cochlear nucleus to the inferior colliculus and auditory cortex. They use different recording methods with Neuropixels in the CN and IC and two-photon calcium imaging in the AC, which may introduce some challenges (described below). Importantly, they use a noise-corrected population-level representational similarity analysis instead of relying on single neuron tuning making the analysis more robust to variability. Conceptually, this approach (likely correctly) supposes that downstream regions incorporate population-level inputs. This approach reveals that some features, like single frequency tuning, are already well defined at the CN, while others, such as amplitude modulation frequency, frequency modulation direction, intensity, and complex sound identity, become decorrelated and more specifically represented at later stages in a feature-dependent manner. Some key findings include that the inferior colliculus is critical for processing AM frequency while the auditory cortex is essential for FM direction and sound identity. These results highlight that auditory features representation across the hierarchy is a mix of gradual and stepwise transformations that enhance the specificity of population-level sound representations. I believe all of my comments can be addressed through textual revisions or relatively modest re-analyses, and I am otherwise supportive of the manuscript's eventual publication.

Some concerns include:

1) Methodological inconsistency across regions. The use of different recording modalities across brain regions, Neuropixels in the cochlear nucleus and inferior colliculus versus two-photon calcium imaging in the auditory cortex introduces potential confounds in comparing representational similarity. In particular, differences in temporal resolution and signal-to-noise characteristics could impact measures of tuning to temporal features such as AM and FM. While the authors' use of a population-level, noise-corrected metric helps mitigate these concerns, matched methodologies across regions would have improved interpretability. This would ideally be addressed with a small number of experiments showing that there are no major differences when using the analytical approaches outlined here (i.e. comparing AC population metrics with calcium imaging versus neuropixels). If that is too difficult, then addressing this in the discussion would also be sufficient. For me, I would want to make sure that these findings are not driven by the chosen recording approach so I would opt for option 1. But it's really up to the authors whether they prefer to pursue additional experimental validation or include a paragraph in the discussion that addresses this concern.

2) Interpretation of decorrelated representations. The central framework of 'decorrelation' as a signature of improved feature representation is well-motivated and clearly supported by the data. However, the study stops short of linking these representational changes to functional outcomes such as perception or behavior. Without this, the claim that decorrelation facilitates categorization or recognition remains speculative. An explicit test of whether more decorrelated representations improve behavioral discrimination or downstream decoding would be required to strengthen the interpretation. This is, of course, not needed for this particular manuscript but could be discussed at greater length in the discussion.

3) Lack of subregion-specific treatment of auditory cortex. A central theme of this manuscript is the hierarchical transformation of sound representations across the auditory pathway. However, the analysis of cortical data treats the AC as a single, unified region, without distinguishing among its known subfields (e.g., A1, AAF, A2, DP). Numerous studies have shown that these subregions differ in their connectivity, temporal dynamics, and selectivity for specific sound features. Some, such as A1 and AAF, are thought to lie earlier in the cortical processing hierarchy, while others, like A2 or DP, are associated with higher-order processing or integrative functions. Collapsing across these subfields obscures potentially important distinctions in how feature representations are transformed and may conflate early cortical representations with those arising later in the hierarchy. If the goal is to map how decorrelation and feature tuning evolve along a sensory hierarchy, then failing to disaggregate auditory cortical subregions undermines the precision of that conclusion. Even if not all subfields could be sampled densely enough for statistical comparisons, it would be important to acknowledge this limitation more directly, or alternatively use relatively standard approaches to determine where the AC recordings are.

Minor

'We trained mice to stay quietly head-fixed' -> How do you know they are quiet? If you don't have microphonic recordings, please edit this to remove 'quietly'.

---

## [Decision Letter · Decision Letter 2]

11 Sep 2025

Dear Dr Bathelier,

Thank you for your patience while we considered your revised manuscript "Feature-dependent decorrelation of sound representations across the auditory pathway" for consideration as a Research Article at PLOS Biology. Your revised study has now been evaluated by the PLOS Biology editors, the Academic Editor and the original reviewers.

In light of the reviews, which you will find at the end of this email, we are pleased to offer you the opportunity to address the remaining points from Reviewer 2 in a revision that we anticipate should not take you very long. We think this is an important point to address.

We will then assess your revised manuscript and your response to the reviewer's comment with our Academic Editor aiming to avoid further rounds of peer-review, although we might need to consult with the reviewers, depending on the nature of the revisions.

In addition to these revisions, we ask you to address a few editorial requests:

* We would like to suggest a different title to improve its accessibility for our broad audience:

Sound feature representations evolve across the mouse auditory pathway

* Please add the links to the funding agencies in the Financial Disclosure statement in the manuscript details.

* DATA POLICY:

Regardless of the method selected, please ensure that you provide the individual numerical values that underlie the summary data displayed in the following figure panels as they are essential for readers to assess your analysis and to reproduce it: 3HI, 4DEF, 5CEF, 6BCH, 7BCE, S1D, S2EIJ, S3EGH, S4DEH, S5BCE and S6BC.

* CODE POLICY

You may also need to complete some formatting changes, which you will receive in a follow up email. A member of our team will be in touch with a set of requests shortly. If you do not receive a separate email within a few days, please assume that checks have been completed, and no additional changes are required.

**IMPORTANT - SUBMITTING YOUR REVISION**

*Resubmission Checklist*

*Published Peer Review*

*PLOS Data Policy*

*Blot and Gel Data Policy*

Sincerely,

Christian

Christian Schnell, PhD

Senior Editor

PLOS Biology

cschnell@plos.org

REVIEWS:

Reviewer #1 (Tetsufumi Ito signed his report): The authors adequately responded to what I mentioned. I do not have any concerns.

Reviewer #2: The authors were generally responsive to the reviews. To address my comment about auditory cortical subfields, they state they have insufficient data to make sub-region analyses, which is fine and understandable. I agree that could be the topic of another study. They do not address, however, where their AC recordings are from in the existing manuscript other than this comment in the reviewer response: 'The AC recordings are mainly sampled from A1 and AAF the primary auditory fields, although not exclusively.' Given the focus of this manuscript, it would appear important to demonstrate this with data in the manuscript itself and not just a comment in the reviewer response. This should be relatively easy given their AC data is collected using 2P imaging and they have the spatial location and tuning of each neuron. Specifically, for each mouse, they can estimate the local tonotopic gradient (plot neural frequency tuning as a function of rostrocaudal location) and empirically confirm whether that particular mouse/recording is from A1, AAF, or otherwise. I leave it to the editor to decide whether this will be needed for final publication. Otherwise, I support publication of this manuscript.

---

## [Editor Report · Decision Letter 3]

3 Oct 2025

Dear Brice,

Thank you for the submission of your revised Research Article "Sound feature representations decorrelate across the mouse auditory pathway" for publication in PLOS Biology. On behalf of my colleagues and the Academic Editor, Manuel Malmierca, I am pleased to say that we can in principle accept your manuscript for publication, provided you address any remaining formatting and reporting issues. These will be detailed in an email you should receive within 2-3 business days from our colleagues in the journal operations team; no action is required from you until then. Please note that we will not be able to formally accept your manuscript and schedule it for publication until you have completed any requested changes.

When you attend to those requests to come, please also make sure to reference the source data in all corresponding figure legends (main manuscript and SI file), for example: 'The source can be found in S1_data' or 'The source data have been uploaded to https://doi.org/10.5281/zenodo.14421103’.

PRESS

Sincerely, 

Christian

Christian Schnell, PhD

Senior Editor

PLOS Biology

cschnell@plos.org